# Development of a Personal Mobility System with Autonomous Driving for Agricultural Work by the Physically Challenged and the Vulnerable

**EunByul Ko [1], KwangHo Han [2] and Chul-Hee Lee [1,*]**

1 Department of Mechanical Engineering, Inha University, 100 Inharo, Michuholgu, Incheon 22212, Korea; eunbyul.avdc@gmail.com
2 Department of Construction Machinery Engineering, Inha University, 100 Inharo, Michuholgu, Incheon 22212, Korea; khhan1736@gmail.com
* Correspondence: chulhee@inha.ac.kr

**Abstract:** Many issues have recently arisen as a result of the aging population and dwindling labor force. To solve these problems, including reduced convenience and productivity, research on the mobility of agricultural machinery and logistics involving autonomous driving is being conducted. However, research on improving individual mobility is scarce. Therefore, an autonomous driving personal mobility system is developed in this study for the physically challenged and other vulnerable groups with mobility difficulties. The personal mobility system can help the physically challenged and the vulnerable to overcome their limited mobility and perform agricultural work. An additional feature is a standing function that allows the performance of tasks too difficult to do while seated. In addition, the autonomous driving function guides the personal mobility system along the work route and stops near agricultural crops. In this study, the design and structural stability of the personal mobility system were analyzed through CAD modeling and structural analysis, with LiDAR applied for autonomous driving and machine learning for agricultural work. This personal mobility system is expected to improve mobility for the physically challenged and the vulnerable, allowing them to play a role and participate more fully in society.

**Keywords:** agricultural work; autonomous driving; LiDAR; machine learning; personal mobility

## 1. Introduction

As the world's population ages, the number of senior citizens grows, and the super-aged society, including the vulnerable and the physically challenged, deepens. Around 901 million people aged 60 and up will live on the planet by 2022, accounting for 12.3% of the global population. By 2030, this will increase to 1.4 billion or 16.5%, and by 2050, it will increase to 2.1 billion, or 21.5% of the global population [1]. Vulnerable groups such as the elderly and the physically challenged have difficulty with indoor and outdoor activities because moving and working independently is inconvenient compared to the general population. They are also vulnerable to both severe and minor mishaps. As a result, advanced countries have been working on initiatives that use assistive technology in various sectors, focusing on personal mobility.

In the agriculture sector, considerable research is being conducted on the work autonomy and autonomous driving of agricultural robots. These robots aim to solve social problems such as the decline in agricultural productivity owing to population aging and the corresponding decrease in human power. In addition, the development of smart mobility systems is crucial to expand the convenience and range of activities of vulnerable groups. Further, the development and demonstration of autonomous wheelchairs are being promoted to improve the convenience of movement for the vulnerable population. In highly functional electric wheelchairs, a controller that implements intelligent algorithms

for learning and judgment is applied instead of a simple feedback controller that uses an existing joystick. Personal mobility is taken for granted by the public, but a personal mobility system is a necessary mode of transportation for the elderly and physically challenged, who are particularly vulnerable. Increased mobility is closely related to improvement in the overall quality of life and affects simple transportation and work and health [2].

Among the various personal mobility systems, the standing wheelchair has medical advantages, such as helping blood circulation and activating the digestive function by lifting the patient who is always sitting into a standing position. A standing position also allows communicating with others at eye level [3]. In addition, the standing wheelchair allows someone to perform tasks that are difficult to do while sitting [4].

Research on autonomous vehicles is constantly underway. Especially since the Fourth Industrial Revolution, the importance of autonomous driving systems has become even greater. Autonomous driving in automobiles is mainly enjoyed by people who are not physically challenged, but the need for technological developments for those who are physically challenged is increasing [5]. Many studies are underway that are focused on autonomous driving for agricultural robots to solve problems in agriculture. In addition, there is development and demonstration of autonomous wheelchairs to improve movement for the vulnerable. For highly functional electric wheelchairs, a controller that implements intelligent algorithms for learning and judgment is applied instead of a simple feedback controller that uses a joystick. Processor performance has greatly improved, with video-processing-based algorithms and the like easily realized [6].

Prassler et al. introduced a robotic wheelchair that provides increased mobility, autonomy, and independence for people with physical restrictions. It has a wide-area navigation control structure to support autonomous driving in a wide and crowded area. Its hierarchical driving structure consists of a basic control level, a tactical level, and a strategic level [7].

In order to drive autonomously to a given destination while avoiding obstacles, an efficient route must be generated. Therefore, it is necessary to generate the shortest route and consider the number of calculations required for route generation, passenger safety, and ride quality. Existing global path generation algorithms, such as Dijkstra's and A *, primarily focus on finding the shortest path. However, Dijkstra's algorithm and A * are slower to calculate, occupy more memory, and are inefficient, compared to other algorithms.

Furthermore, with the A * algorithm, a route cannot be generated to deal with sudden changes, so it is not suitable for high-speed driving. The dynamic window approach is a regional route generation and obstacle avoidance algorithm that can select the most appropriate speed. It produces a smoother path than other methods but has a disadvantage in that it cannot easily reach a locally optimal solution. To make up for these shortcomings, and a mix of algorithms is often used, rather than writing just one [8].

In this study on the personal mobility system, a standing function was added to an electric wheelchair to develop a personal mobility system for the physically challenged and the vulnerable. Its structural stability was also analyzed through structural analysis. Convenience functions such as standing and rotating were applied. In addition, autonomous driving was applied to smart mobility systems such as wheelchairs for the convenience of the agricultural population, and deep learning technology using riders and vision cameras was utilized for route tracking and image processing. This study followed a campus map for the experiment's progress and used tomatoes as the fruit tracked to recreate a rural environment.

This research makes it possible to improve the mobility of the physically challenged and vulnerable, who find it inconvenient to carry out agricultural work. It is also expected that participating in agricultural work more easily can help them play a meaningful role as a member of society.

## 2. Design and Analysis

### 2.1. Design

The personal mobility system developed in this paper was designed with reference to the existing electric wheelchair [4]. A front-wheel-drive system was used. The rear wheels are a caster system, left and right front wheels are firewood, and space for the motor is secured through a bevel gear. Steering is possible by using a different RPM for each motor, and the standing function is implemented with an electric actuator. The kinematic structure for the standing function is composed of a four-bar link so the angle between the backrest and the seat can be increased. CAD modeling of the personal mobility system is shown in Figure 1. Acceleration and constant velocity torque were calculated, and an appropriate motor was selected. To calculate the torque at a maximum speed of 5 km/h, the weight (m), diameter of the drive wheel (D), RPM, and acceleration time (t) are required (summarized in Table 1). Weight is the sum of the weight of the personal mobility system (approximately 50 kg) and the maximum user weight (100 kg). For the load moment of inertia, Equation (1) is used, and the acceleration and constant velocity torques can be obtained using Equations (2) and (3), respectively. Therefore, the required motor torque is the sum of the acceleration and constant velocity torques and can be obtained at 308.454 kg·cm by applying a safety factor of 1.5. As a result, a motor with 3000 RPM, 11.2 kg·cm, and 1/30 reducer was selected.

$$J = m \cdot D^2 / 8 \tag{1}$$

$$T_a = \frac{J \cdot 2\pi \cdot RPM}{60 \cdot g \cdot t} \tag{2}$$

$$T_m = \mu \cdot m \cdot D / 4 \tag{3}$$

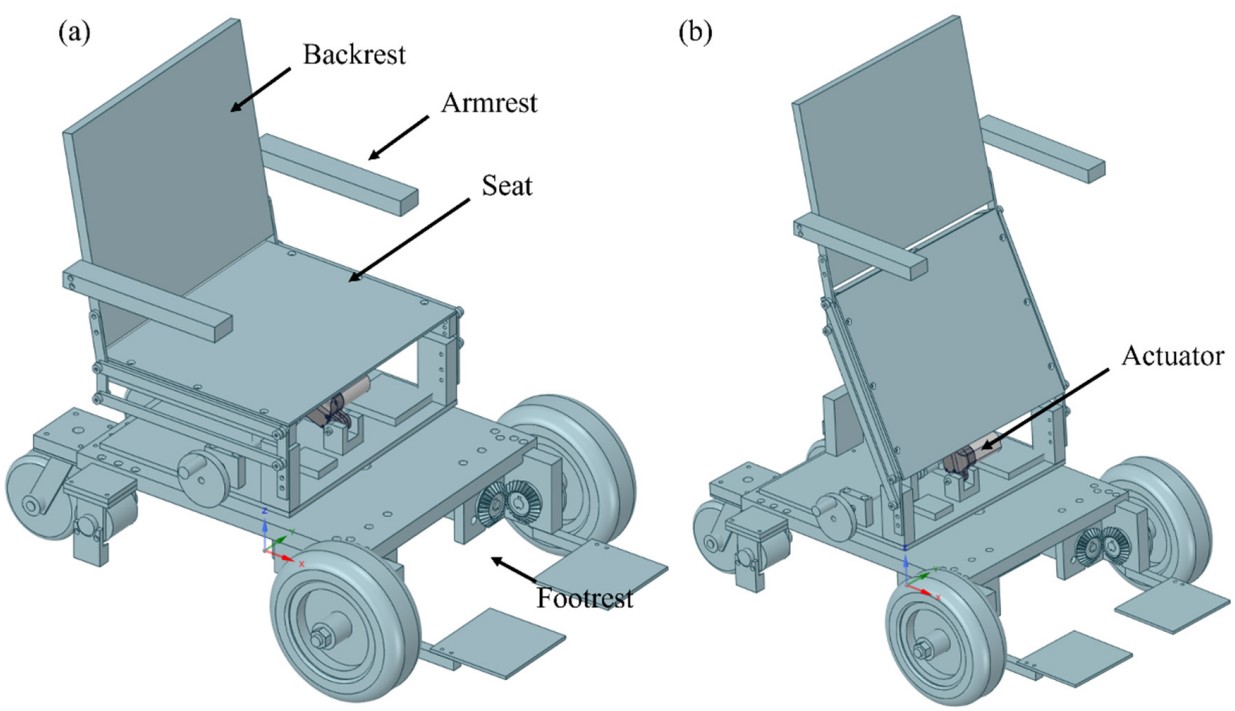

**Figure 1.** CAD modeling of the personal mobility system (**a**) driving and (**b**) standing modes.

**Table 1.** Parameters required to calculate torque at a maximum speed of 5 km/h.

| Parameter | Value |
|---|---|
| Weight<br>(Sum of personal mobility system and maximum user weight) | 150 kg |
| Diameter of the driving wheel | 28.21 cm |
| *Maximum speed* | 94.03 rpm |
| Acceleration time | 1.5 s |

### 2.2. Analysis

After designing the personal mobility system, structural analysis was performed to examine the structural stability with someone aboard. In order to shorten the analysis time, parts not required for structural analysis were excluded but are still shown in Figure 2.

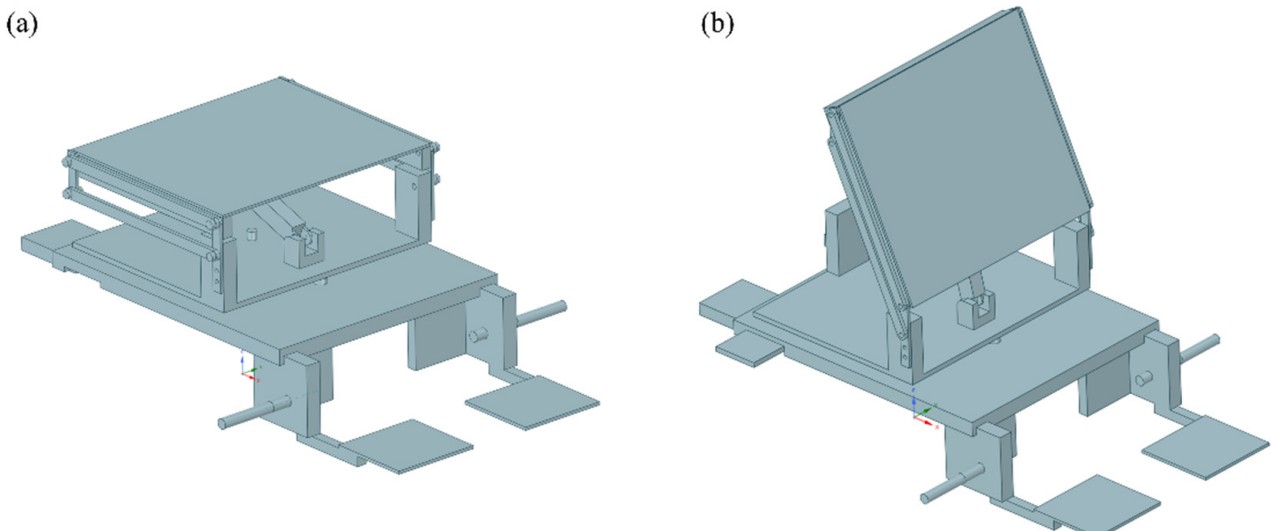

**Figure 2.** Modeling of the personal mobility system for structural analysis: (**a**) seating mode and (**b**) standing mode.

The materials for the personal mobility system developed in this study, detailed in Table 2, were aluminum 5052 and MC nylon. MC nylon was used for the seat, and aluminum 5052 was used for the rest of the parts. According to the analysis results, aluminum 5052 was more suitable than MC nylon and was used. For boundary conditions, a fixed support was applied to the front wheel shaft and the upper surface of the rear wheel casters, with 1000N applied to the seat.

**Table 2.** Properties of the materials.

| | Aluminum 5052 | MC Nylon |
|---|---|---|
| Density | 2680 kg/m$^3$ | 1160 kg/m$^3$ |
| Young's modulus | 70.3 GPa | 3.432 GPa |
| Poisson's ratio | 0.33 | 0.4 |
| Yield strength | 193 MPa | 96 MPa |

Figure 3 shows the boundary condition, Figure 3a presents the condition for force, and Figure 3b shows the fixed condition. The analysis results for seating and standing modes are shown in Figure 4a,b, respectively. From checking the maximum equivalent stress, the seating mode is 29.584 MPa in the four-bar link, and standing mode is 75.875 MPa for the footrest. This is lower than the 193 MPa yield strength of aluminum 5052 and satisfies

the safety factors of 6.52 and 2.54. The deformation and stress values are summarized in Table 3.

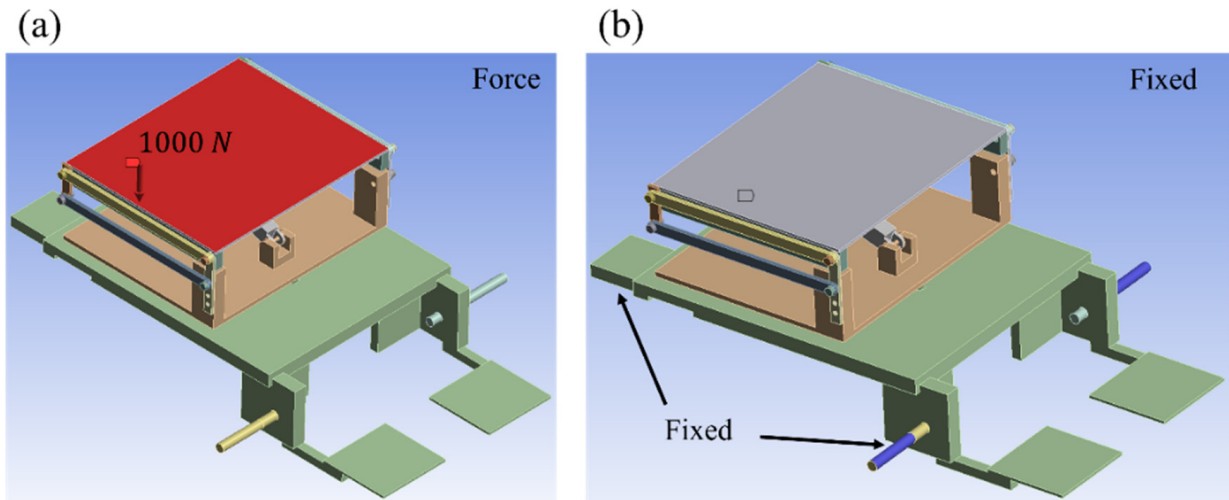

**Figure 3.** Boundary conditions: (**a**) force (**b**) fixed.

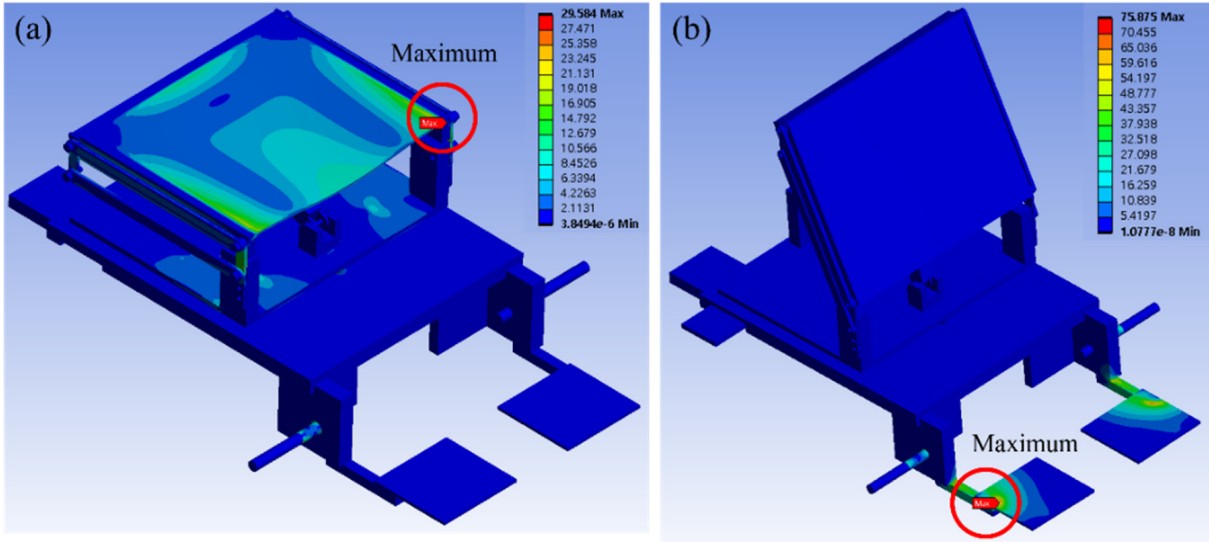

**Figure 4.** Structural analysis results: (**a**) seating and (**b**) standing modes.

**Table 3.** Deformation and stress values from structural analysis.

|  | **Seating Mode** | **Standing Mode** |
|---|---|---|
| Deformation | 14.923 mm | 3.413 mm |
| Stress | 29.584 MPa | 75.875 MPa |
| Safety factor | 6.52 | 2.54 |

## 3. System Overview

### 3.1. Hardware

The motor and battery were attached to the lower end of the prototype so it would not be difficult to move. The configuration and wiring attached to the lower end of the prototype is shown in Figure 5. The motor, motor driver, and battery are mounted at the bottom to drive the prototype. The attachment of the lower end is shown in Figure 6. The two motors are attached to the battery and the embedded board via the motor driver. A driver for control was installed according to specifications, and a lithium-ion battery

supplies power to the two motors. A cushion-shaped chair was attached to give a sense of stability to the prototype, and a device for autonomous driving was added, as shown in Figure 7 and described in Section 3.2. A prototype based on the design was prepared for experimental verification and is shown in Figure 7. This study calculated the acceleration and constant velocity torque for the drive motor and used a BLDC motor equipped with a reducer, as described in Section 2.1.

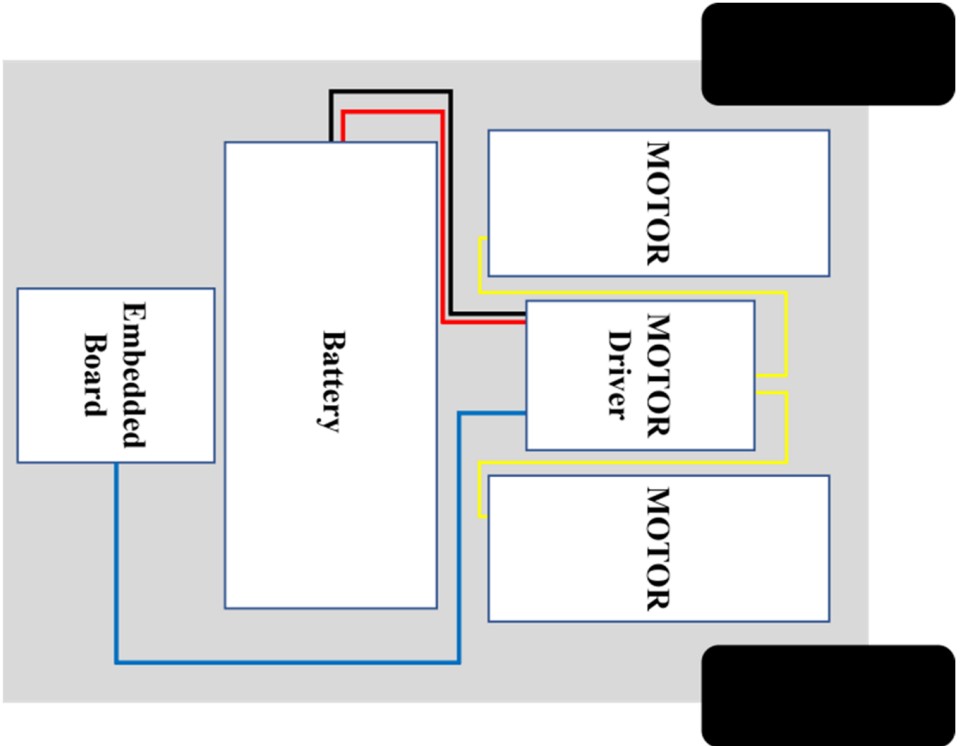

**Figure 5.** Configuration and wiring attached to the lower end of the prototype.

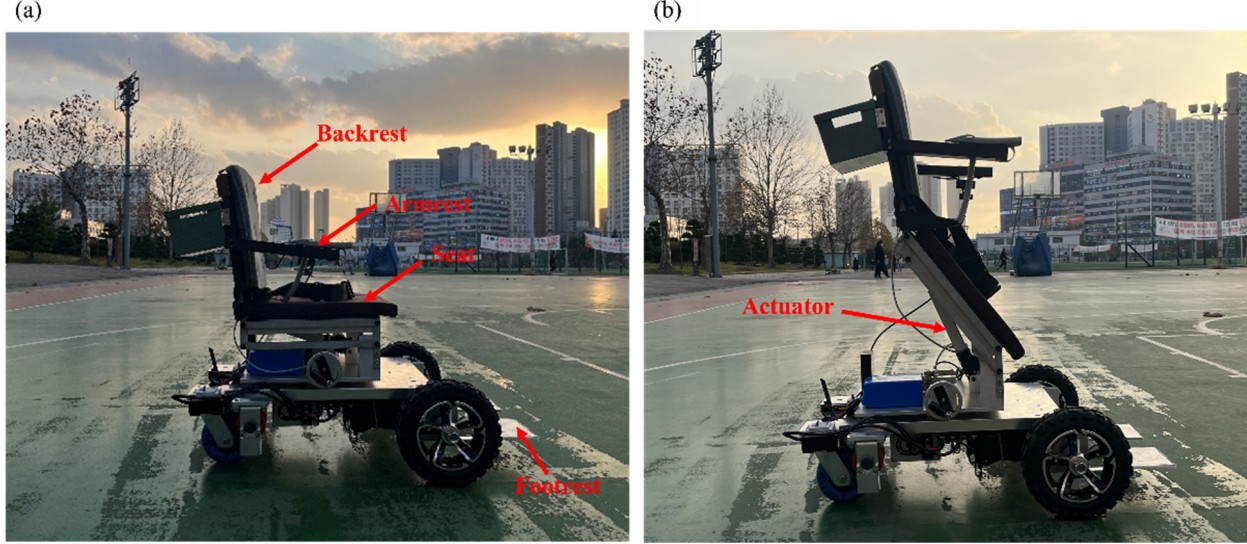

**Figure 6.** Prototype of the personal mobility system: (**a**) seating and (**b**) standing modes.

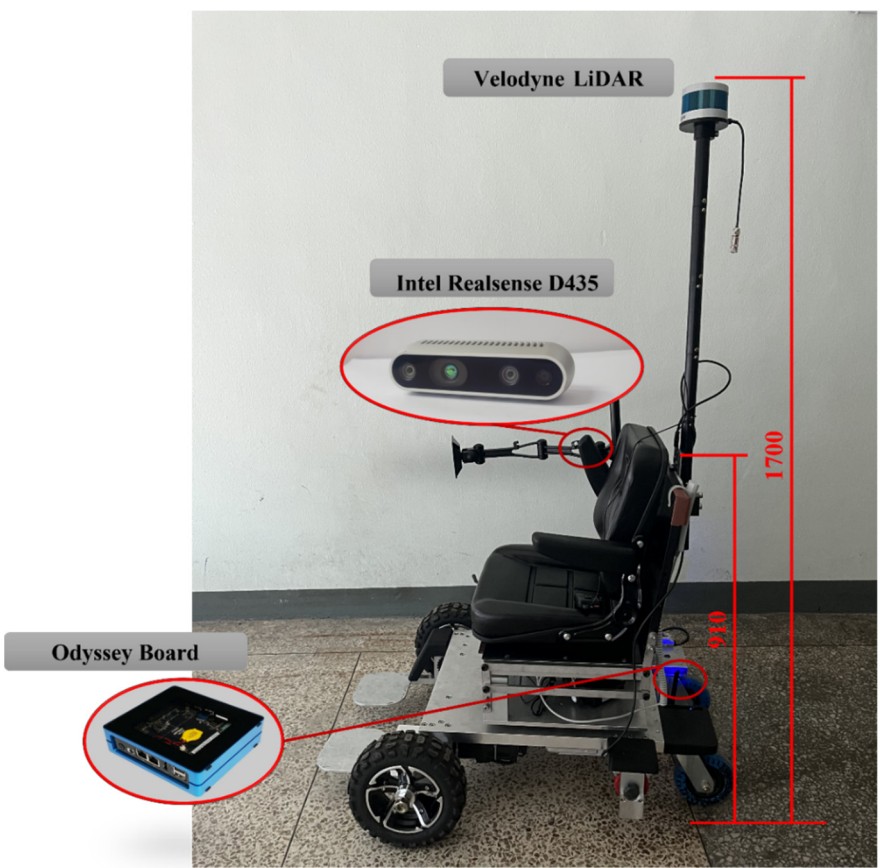

**Figure 7.** Prototype showing the mounting positions of the camera and LiDAR sensor. Distance data in mm.

### *3.2. Software*

The software used for driving was the open-source ROS (Robot Operating System) melodic version of the Ubuntu 18.04 operating system. The MD motor drivers and packages were used to drive the motor with ROS. First, the ROSCORE instruction was used to execute the master such that connections and message communication between nodes could be performed. The Velodyne LiDAR package was used to receive LiDAR point cloud values. The utilized CPU was an Intel Core i7-7700HQ, 2.8 GHz, with the NVIDIA GTX1050Ti GPU. The dataset used and learned in YOLO v3 included approximately 550 screened tomato images. A modeling-based prototype, shown in Figure 6, was designed for the experiment. The sensors for autonomous driving were attached to the prototype, and their heights and positions are depicted in Figure 7. As shown in Figure 7, the LiDAR was mounted at a height of 1700 mm, and the vision camera was mounted on the armrest for storage. An Odyssey board was attached to the seat. Simultaneous localization and mapping (SLAM) is a method that generally uses various sensors to create a map of the surrounding environment to estimate one's position [9]. The master is run using roscore for node-to-node connections and communications. To receive LiDAR point cloud values, this study used the Velodyne LiDAR package to receive topics and used the advanced implementation of Lidar Odometry and Mapping (A-LOAM) to recognize the surrounding environment. LOAM is used to analyze the surroundings simultaneously with position estimation [10].

### *3.3. Deep-Learning Algorithm for the Autonomous System*

The RRT and A* algorithms for autonomous driving are described in this section. This study determines the shortest path between the beginning and goal places without colliding with any obstacles in order to discover and design the best route. In this investigation, an

algorithm was chosen and used. The following is a quick summary of the two algorithms. The A* algorithm first calculates the shortest path from the initial node (beginning point) to the destination node (target point). The operation method of the A* algorithm used in the experiment is shown in Figure 8. The RRT method is a sampling-based path-planning technique that creates points at random locations throughout the state space and utilizes this as a propensity to expand the tree quickly from the beginning point to generate a path that leads to the destination.

---

*OPEN : start point*

*CLOSED : set param*

*while OPEN.top( ) != Goal:*

　　　*current = OPEN.top( ).second: OPEN.pop( ):*

　　　*CLOSED.insert($n_r$)*

　　　*for $n_{neighbor}$ in adj [$n_r$]:*

　　　　　*cost = g[$n_r$] + cost($n_r$, $n_{neighbor}$)*

　　　　　*if $n_{neighbor}$ in OPEN and cost < g[$n_{neighbor}$]:*

　　　　　　　*remove $n_{neighbor}$ from OPEN*

　　　　　*if $n_{neighbor}$ in CLOSED and cost < g[$n_{neighbor}$]:*

　　　　　　　*remove $n_{neighbor}$ from CLOSED*

　　　　　*if neighbor not in OPEN and neighbor not in CLOSED:*

　　　　　　　*g[neighbor] = cost*

　　　　　　　*OPEN.push({f[$n_{neighbor}$], $n_{neighbor}$})*

---

**Figure 8.** Operation method of the A* algorithm.

The A-LOAM algorithm using a LiDAR was created. At this time, if there is an obstacle on the route over which the personal mobility system travels, a route must be generated around the obstacle. The RRT algorithm and the A* algorithm was compared, and the final algorithm to be used was selected. The A* algorithm is the least costly and looks at the nodes adjacent to a particular node in the state space and goes from the starting point to the target point. At each iteration of its main loop, A* needs to determine which of its paths to extend. It does so based on the cost of the path and an estimate of the cost required to extend the path all the way to the goal. Specifically, A* selects the path that minimizes.

$$f(n) = g(n) + h(n) \tag{4}$$

$h(n)$ is a heuristic function that estimates the cost of the cheapest path from n to the objective, where $n$ is the next node on the path, $g(n)$ is the cost of the path from the start node to $n$, and $g(n)$ is the cost of the path from the start node to $n$. The RRT method is far faster than the A* algorithm; nevertheless, the RRT algorithm's computed path is significantly longer than the A* algorithm, which is a significant drawback. The effects of such distance measuring technologies are the subject of ongoing research. Velodyne LiDAR data was translated into 2D laser data using an algorithm to create a map, and trials were undertaken on straight roadways to test the reaction speed. The RRT [11] method first locates a configuration space, then enters an initial configuration state ($q_{init}$) and a goal configuration state ($q_{goal}$), selecting any feasible q from q init and connecting them with edges. The technique is then continued until the $q$ target is attained, at which point the best path is created. The operation method of the RRT algorithm used in the experiment

is shown in Figure 9 [12]. An experiment was conducted to evaluate the performance of the prototype in an environment with trees. Ripe tomatoes and unprocessed tomatoes were attached to trees to construct an experimental environment. To distinguish tomatoes, the vision camera uses Intel's RealSense D435 (a 3D camera). After driving the prototype, YOLO v3-Tiny used the learned weight file to detect harvestable tomatoes in an image, distinguishing between harvestable (red and ripe) tomatoes and unripen (green) tomatoes. This system extracted distance information from the center point of the tomato's boundary box, as detected by the 3D camera. Object recognition and detection was performed using deep learning. The deep learning algorithm used is YOLO v3, plus a deep neural network learning layer and multilayer perceptron (MLP) consisting of an output layer structure and multiple hidden layers [13–15]. YOLO v3 has been applied to various types of algorithms such as the recurrent neural network, although YOLO v4 and YOLO v5 were recently released [16,17]. The structure of YOLO is like a region-based convolutional neural network (R-CNN) in that it consists of bounding box classification, positioning, deduplication, and post-processing, but it defines a series of object detection procedures as one regression problem [18,19]. Although these one-stage-detection systems reduce the accuracy, the detection speed (in frames per second) is greatly increased so that the object can be recognized in real time at a rapid detection speed. This has evoked a large wave of cognitive deep learning. According to the trends in artificial-intelligence-related research for the visually impaired, YOLO showed the highest number of uses at 2.7% as an application technology for domestic research design and development implementation, and YOLO v2 showed the accuracy and speed of an R-CNN [20]. The improved Fast R-CNN and the Faster R-CNN showed higher accuracy and speed [21,22]. From the third version of YOLO used in various projects, this study selected and used the Tiny version, which has a slightly lower accuracy but a relatively small memory capacity and a high detection speed.

---

*Input* : $x_{init}$ , $X_{goal}$

*Output* : Tree T = (V,E)

$V \leftarrow x_{init}$ , $E \leftarrow \emptyset$ ;

**for** *i = 1, ..., N* **do**

    $x_{rand}$ ← *RandomSample (i)* ;

    $x_{nearest}$ ← *Nearest (T , $x_{rand}$)* ;

    $x_{new}$ ← *Steer ($x_{nearest}$ , $x_{rand}$)* ;

    **if** *ObstacleFree ($x_{nearest}$ , $x_{new}$)* **then**

        *V* ← *V* ∪ {$x_{new}$};

        $X_{near}$ ← *Near (T, $x_{new}$ , r);*

        *ChooseParent ($X_{near}$ , $x_{nearest}$ , $x_{new}$ , E);*

        *Rewiring ($X_{near}$ , $x_{new}$ , E);*

    **end**

**end**

---

**Figure 9.** Operation method of the RRT algorithm.

The convolutional network process in Figure 10 populates weight data with the image. Therefore, a pre-trained weight file is needed for the target object. In YOLO training, the larger the number of images in the dataset for each class (the target objects to be detected), and the more diverse the angle and lighting environment, the more accurate the detection will be in various actual environments. In this study, 550 images out of about 900 images from Kaggle's free tomato dataset, which is a machine learning platform, were selected. To use the selected dataset for training, a bounding box is drawn around the target object to be detected from each image.

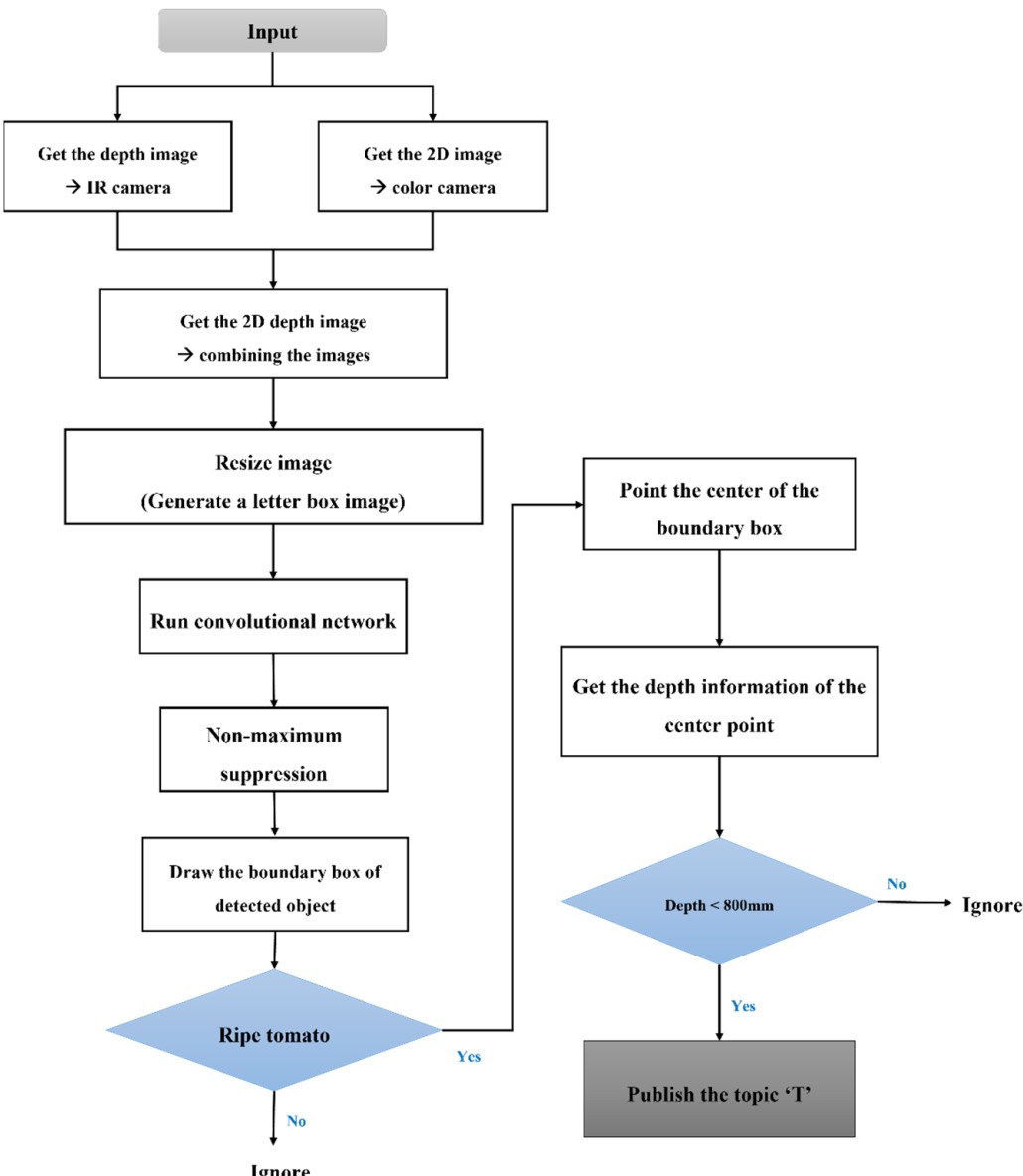

**Figure 10.** Flowchart of the crop recognition system.

*3.4. Experiment*

In this study, an algorithm that constructed a map using LiDAR and reached the target point was used for experimentation. First, a map of the location where the occupant wants to drive was created through mapping. The Velodyne data were changed to a two-dimensional (2D) laser for generating the map. To apply each algorithm and select the optimum algorithm, in the experiment, obstacles were set using safety cones on the campus and the mean of the values for traveling 10 times and reaction speed at each position were

obtained. First, the state of point cloud recognition before and after filtering by the rider appears, as shown in Figure 9. Point clouds and maps can be viewed through RVIZ, a visualization tool for ROS, and obstacles, trees, and human shapes can be observed through the point cloud. In addition, to derive these data values, and the values of the results of the experiments with the algorithm applied are listed in Table 4.

**Table 4.** Algorithm reaction speeds during driving (①, ②, ③). The route environment of each symbol is shown in Figure 12.

|  | Position ① | Position ② | Position ③ |
| --- | --- | --- | --- |
| A* | 1.368 (s) | 0.478 (s) | 1.962 (s) |
| RRT | 1.498 (s) | 0.871 (s) | 2.689 (s) |

Figure 11 shows the tomato recognition in the experimental environment. Table 5 lists the recognition results of the final experiment for the tomato positions. The system did not recognize unripe tomatoes, and the recognition rate of the last tomatoes was approximately 40%. It was confirmed that other crops can be easily processed for recognition. For this, the system was configured to detect harvestable fruit. The results of running in an ROS environment using the tomato weight file learned for the experiment indicated that when the harvestable tomato distance is 600 mm (depth 800 mm), topic 'T' is published to the subscriber node, and when 1 m (depth 800 mm), the message is not published.

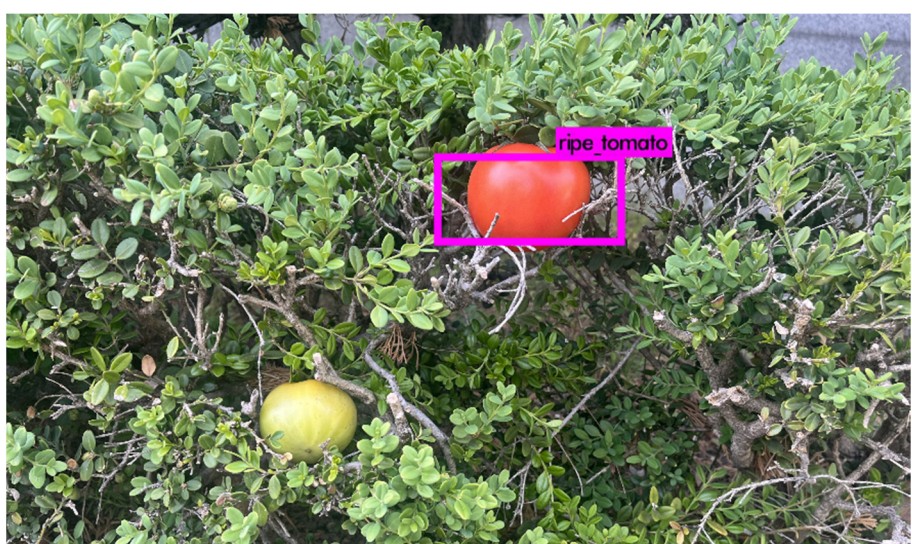

**Figure 11.** Tomato recognition.

**Table 5.** The structural analysis results; T: tomatoes located in Figure 8c.

| Recognition (O) | T1 | T2 | T3 | T4 | T5 | T6 | T7 | T8 |
| --- | --- | --- | --- | --- | --- | --- | --- | --- |
| **TEST 1** | O | - | - | O | - | - | O | O |
| **TEST 2** | O | - | - | O | - | - | - | O |
| **TEST 3** | O | - | - | O | - | - | - | O |
| **TEST 4** | O | - | - | O | - | - | - | O |
| **TEST 5** | O | - | - | O | - | - | O | O |

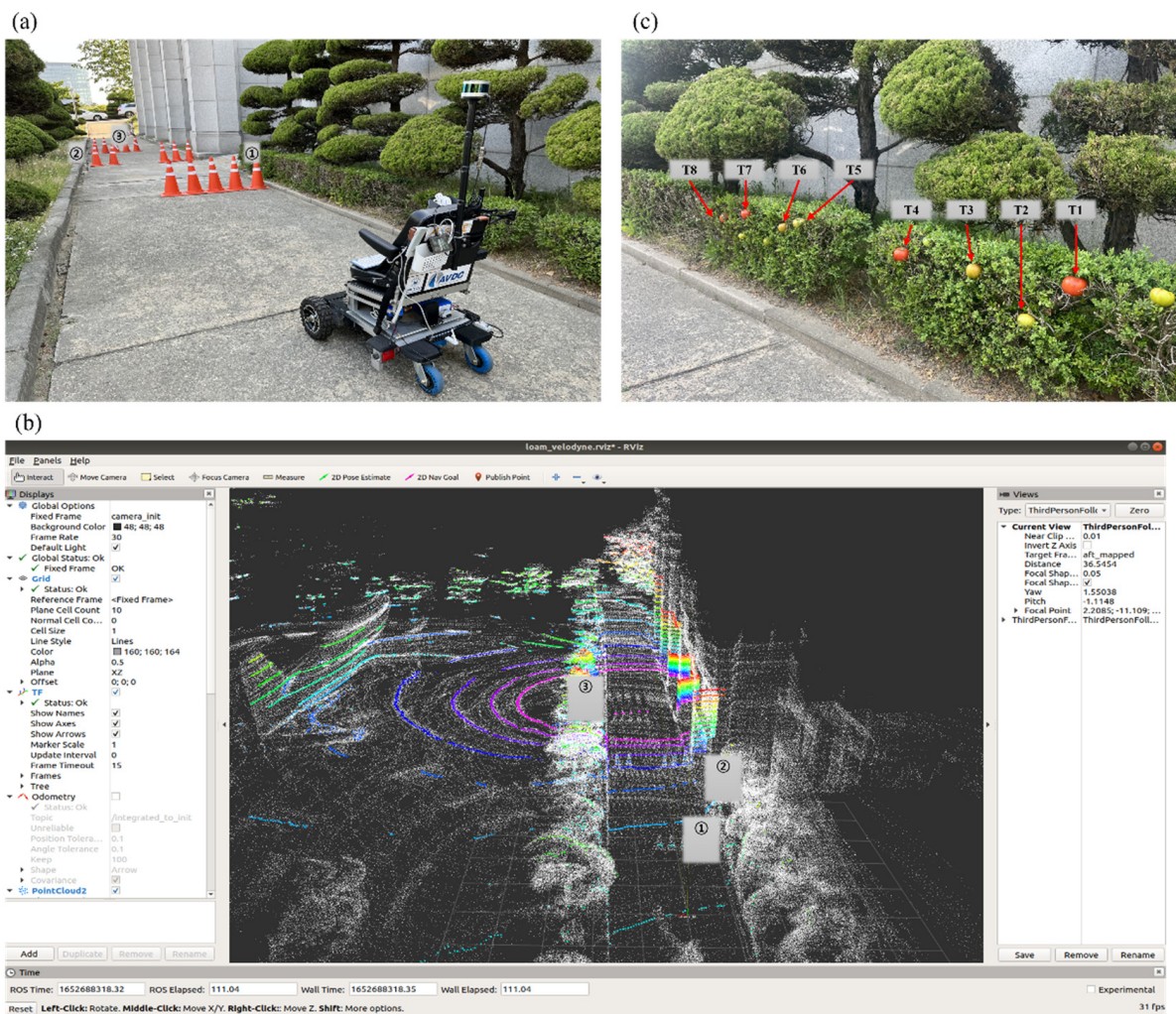

**Figure 12.** Establishment of the autonomous driving test environment: (**a**) experimental environment; (**b**) 3D map collected point cloud; (**c**) experimental environment for tomato recognition.

## 4. Conclusions

In this study, a personal mobility system was developed for the vulnerable and physically challenged, and the stability of the system was examined through structural analysis. The results revealed that the safety factors of 6.52 and 2.54 were met, and that stability was not a concern even when a user weighing up to 100 kg was onboard. The RRT algorithm is much quicker than the A* method, although the RRT algorithm's computed path is longer than the A* algorithm. Furthermore, because the distance measuring method utilized affects these algorithms, they have been regularly explored (e.g., Manhattan and Euclidean distances). In YOLO, 550 datasets are usually adequate for object recognition, rather than hundreds of photos for a single class. The system did not recognize unripe tomatoes, and the recognition rate of the last tomatoes was approximately 40%. It was confirmed that other crops can be easily processed for recognition. For this, the system was configured to detect harvestable fruit. As a result, depending on the angle, illumination, and type of item, a poor identification rate was seen in repeated testing. Supplementary training utilizing high-quality data with excellent picture quality in a real agricultural context can enhance the identification rate. The use of machine learning techniques by increasing the dataset should be included in this thesis, and more experiments with route-running experiment data are necessary. In this study, Velodyne LiDAR is primarily utilized for route driving and obstacle avoidance. Sensor fusion is largely employed in self-driving and self-working vehicles. However, in the case of a vision camera, an inaccuracy in

the identification rate arises owing to the light and angle during sensor fusion. Future research must use Velodyne LiDAR and make use of the variation in reflectance between the surfaces of fruits and leaves to compensate for the identification rate issue. Using Velodyne LiDAR in Sweden, an algorithm based on apple reflectivity was created. It is conceivable to operate a personal mobility system in an agricultural area with one sensor and no sensor fusion if a fruit recognition algorithm employing Velodyne LiDAR is applied. Therefore, in future research the aim is to develop an algorithm that can apply autonomous driving and autonomous work using Velodyne LiDAR. Algorithm development using one sensor is expected to contribute to the development of more convenient and efficient smart mobility systems.

**Author Contributions:** Conceptualization, E.K. and K.H.; methodology, E.K.; software, E.K.; validation, E.K. and K.H.; formal analysis, E.K.; investigation, E.K.; resources, K.H.; data curation, E.K.; writing—original draft preparation, E.K.; writing—review and editing, C.-H.L.; visualization, E.K.; supervision, C.-H.L.; project administration, C.-H.L.; funding acquisition, C.-H.L. All authors have read and agreed to the published version of the manuscript.

**Funding:** This research received no external funding.

**Institutional Review Board Statement:** Not applicable.

**Informed Consent Statement:** Not applicable.

**Data Availability Statement:** Not applicable.

**Acknowledgments:** This work was carried out with the support of "Cooperative Research Program for Agriculture Science and Technology Development (Project No. PJ015312022022)" Rural Development Administration, Republic of Korea.

**Conflicts of Interest:** The authors declare no conflict of interest.

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
