# Peer review of "Development of a Personal Mobility System with Autonomous Driving for Agricultural Work by the Physically Challenged and the Vulnerable"

_agriculture, doi:10.3390/agriculture12071054_

Round 1

Reviewer 1 Report

The authors are addressing a highly relevant topic by developing a mobility concept for physically challenged people within the agricultural work environment

The description of the mobility device within the abstract is a bit too generic. At least a concrete field of application for the mobility device should be addressed in order to clarify the focus already from the beginning of the article. 

It remains unclear if the authors from line 36 on are addressing the needs to a wheelchair or if they are already describing the advantages of their concept, (I assume the first option but this should be clarified in the text). The following focus on needs for autonomous mobility systems within the agricultural sector seems confusing. From the title, the reader expects mobility devices for handicapped people working in the agricultural sector but I think this is not what the authors intend to tell. What is the real use case?

The description of the design is superficial and purely descriptive. The authors are claiming to rely on an existing concept but there does not seem to be a scientific design process underlying the development. If so than it should be clearly described in the paper.

The system overview including hardware, software and the learning algorithm is descriptive but well structured. The analysis of the included experiment is purely descriptive without any statistical analysis.

I really appreciate your concept and the development effort but to be published the paper needs strong rework.

Author Response

That is an interesting query.
To help you understand, I've added a note on research objectives to clarify the focus on mobile devices.
Line 36 explains the advantages of the standing function. 
If you sit first, it helps with difficult activities. 
For example, if you make a person standing like a normal person through the standing function, it can help blood circulation and digestion. 
In addition, in the case of a disabled person using a wheelchair, it is difficult to keep eye level with a standing person because they are only sitting.

If you solve this problem, you can help with housework or work environment. 
In fact, the purpose of this study is to harvest fruits in a high position in the agricultural environment.

Second, we detailed the design for greater stability and expanded the considerations of materials and loads. 
We have provided a process that can be safer for users through analysis. 
The content of the design claims that it can be conveniently used by users through standing and various other convenient functions.

We agree with you and have incorporated this suggestion throughout our paper. 
I checked your review and added a statistical addition to the result and explained how to improve it.
Statistical values ​​for the tomato recognition experiment were expressed as data.
The system did not recognize unripe tomatoes, and the recognition rate of the last tomatoes was approximately 40%. 
Confirmed that other crops can be easily processed for recognition. 
For this, the system was configured to detect harvestable fruit. 
And for the part where the recognition rate was low, I wrote that it was due to the lack of data learning. 
As a supplement to future research, more than 550 tomato images will be trained, and this is an area that needs to be sufficiently supplemented in the research. 
Thank you for providing these insights.

Reviewer 2 Report

This is a very meaningful work can help farmers efficiently complete agricultural labor. However, I still have the following suggestions:

1. The analysis is not sufficient. There are too few references.

2. There are spelling mistakes in the first sentence in Abstract, and the language expression needs to be improved carefully.

Author Response

First of all, thank you for writing a review for my paper.
Write a reply to your review.

Coments 1: There will be many studies in a research field similar to mine.
During the course of the study, I consulted papers on important points.
However, I have referred to the necessary parts in writing the paper and conducting the experiment, and I think it is not lacking.

Coments 2 : We agree with your assessment. Spelling errors in the abstract were my mistake and have been corrected.
Other spelling errors are also being checked and corrected, and confirmed errors are corrected first.
Thanks for commenting on my mistake.

Round 2

Reviewer 1 Report

Thank you for your re-comments and improvements. Still i fear that one of my main concerns haven´t been addressed: The  focus on needs for autonomous mobility systems within the agricultural sector seems confusing. From the title, the reader expects mobility devices for handicapped people working in the agricultural sector but in your description the needs for such devices for elderly and vulnerable peope is more described from a medical/caring perspective not from a working perspective in the agricultural sector. Is it about the needs and the related solution for elderly/vulnerable people in general? Or is it about a device for working in the agricultural sector by physically challenged (like in the title)? For this journal the second would be necessary - but then you´ll have to adress the needs and requirements for physically challenged working in the agricultural sectur (from line 37 on).

Author Response

I understand what you are concerned about.
However, looking at the agricultural environment, there are more problems due to aging than young people.
I think of this problem as follows.
A lot of young people come to the city, so there are a lot of old people.
I thought of it as a problem.
But, as you said, my thesis lacks this part.
I will add and edit this part.
Thanks for pointing it out.
